# Seasonal and Spatial Distribution of Fall Armyworm Larvae in Maize Fields: Implications for Integrated Pest Management

**DOI:** 10.3390/insects16020145

**Published:** 2025-02-01

**Authors:** Karimou Zanzana, Antonio Sinzogan, Ghislain T. Tepa-Yotto, Elie Dannon, Georg Goergen, Manuele Tamò

**Affiliations:** 1Laboratoire d’Entomologie Agricole (LEAg), Faculté des Sciences Agronomiques (FSA), Université d’Abomey-Calavi (UAC), Abomey-Calavi, Cotonou 01 B.P. 526, Benin; sinzogan2001@yahoo.fr; 2Biorisk Management Facility (BIMAF), International Institute of Tropical Agriculture (IITA-Benin), Cotonou 08 B.P. 0932, Benin; g.goergen@cgiar.org (G.G.); m.tamo@cgiar.org (M.T.); 3Ecole de Gestion et de Production Végétale et Semencière (EGPVS), Université Nationale d’Agriculture (UNA), Kétou B.P. 43, Benin; 4Laboratoire des Sciences Naturelles et Applications (LNSA), Ecole Normale Supérieure de Natitingou, Université Nationale des Sciences, Technologies, Ingénieries, et de Mathématiques (UNSTIM), Abomey B.P. 486, Benin; edannon@gmail.com

**Keywords:** climate variability, spatiotemporal distribution, damage, larval dispersion

## Abstract

The fall armyworm is a destructive pest that severely impacts maize crops. This study focuses on understanding how this pest spreads in maize fields and how the population fluctuates during both the dry and rainy seasons. We examined maize plants in two different agroecological regions (zones 6 and 8) of southern Benin, tracking the number of larvae, the level of infestation, and the extent of plant damage. Higher larval infestation and damage were observed in zone 8 (fisheries region) during the dry season compared to zone 6 (ferralitic soils region). However, during the rainy season, while more plants were damaged in zone 8, the number of larvae was similar between the two areas. Fall armyworms tend to form small colonies when laying eggs. This behavior influences how they spread and attack crops.

## 1. Introduction

In the context of mounting concerns over climate change, the global agricultural landscape is facing a new major challenge, the fall armyworm (FAW), *Spodoptera frugiperda* J E Smith, (Lepidoptera: Noctuidae). Native to the tropical and subtropical regions of the Americas, this invasive insect pest has rapidly become a great threat to agriculture in Africa [1] and Asia [2,3]. Its voracious larvae and rapid spread across regions have induced heavy damage to several crops such as sorghum and maize [4,5,6]. Estimated annual yield losses due to FAW in African countries range from 8.3 to 20.6 million tons, with losses reaching up to 735,000 tons in Benin [7]. Beyond its voracity, FAW is a polyphagous insect species, infesting over 353 host plant species, with a clear preference for maize [8]. The combination of its remarkable morphological plasticity and polyphagous feeding habits [9,10] is posing a serious threat to agricultural productivity and food security [11], particularly in developing countries [11,12,13,14].

In southern Benin, maize is cultivated during both the rainy and dry seasons and is the major cereal crop in AEZ 6 and 8 (see below detailed description), justifying the choice of these zones for the study. The rainy season extends from April to July, while the dry season spans from November to March. During the dry season, maize cultivation is primarily limited to irrigated areas or valleys with higher water availability. Furthermore, FAW infestations are prevalent in both seasons across AEZ 6 and 8, impacting crop health and yield. To manage FAW, farmers often rely on synthetic pesticides [15]. While these pesticides are known for their effectiveness, economic constraints restrict their use to a limited number of farmers, as not all can afford the costs associated with regular applications.

For a better understanding of the bio-ecology of this invasive pest as the basis for developing sustainable management strategies, the current study focuses on the seasonal fluctuations of its larval population density and the factors that affect its spatial dispersion [16,17]. These fluctuations are driven by a combination of ecological and agroclimatic factors, such as competition, natural enemies, resource availability, temperature, precipitation, and relative humidity [18,19,20,21]. Additionally, the rate of change in the FAW population over time is influenced by its fecundity, speed of development, and survival. Environmental and numerous biotic factors shape all these variables [17,22].

The spatial distribution pattern is an intrinsic factor of species and may reflect their behavioral traits due to the interaction between insects and their environment [23,24]. Thus, effective integrated pest management tactics can benefit from investigations of spatial dispersion data. Several studies have attempted to characterize FAW spatial distribution on maize in Mexico [25,26,27], and in China [28], documenting various distribution patterns for these pest populations. However, the spatial distribution of FAW in maize fields has not yet been investigated in Africa. Aggregation indexes and distribution frequencies can infer distribution types [29,30]. While non-mathematical, these indexes provide an approximate description of population distributions. Based on observed field data, probability distribution models are crucial for establishing sampling and statistical analysis criteria for pest management [27,31].

This study attempts to elucidate the spatiotemporal distribution of FAW associated with the larvae in maize fields in Benin, where maize is one of the staple food security crops. It capitalizes on recent findings by [22,32,33], whereby temperature, relative humidity, and rainfall were considered important agroclimatic factors affecting FAW populations [34].

## 2. Material and Methods

### 2.1. Study Area

This study was carried out in the southern region of Benin, more specifically in the two agroecological zones, AZE 6 and AEZ 8 [35]. Zone 6 (AEZ 6) “called zone of terre de barre” (Southern and Central Benin), is characterized by the presence of ferralitic soils, a Sudano-Guinean climate (two rainy seasons alternating with two dry seasons) with an annual rainfall of 800–1400 mm per year and an altitude range of 20 to 250 m, while zone 8 (AEZ 8), called “fisheries region” (Southern Benin), is dominated by coastal gleysols and arenosols with a Sudano-Guinean climate and annual rainfall of 900–1400 mm per year and an altitude range of 0 and 130 m. With a subequatorial climate featuring two rainy (April–July and September–November) and two dry seasons (December–March and August–September) and a growing season of about 240 days, the area encompassing AEZs 6 and 8 spans over 17,920 km^2^. Fisheries and crop production, with a particular emphasis on maize, are the main drivers of this region’s economy. Sampling in AEZ 8 occurred in the municipalities of Adjohoun (6°42′43″ N/2°29′3″ E), Dangbo (6°34′4″ N/2°29′57″ E), Ouinhi (7°4′5″ N/2° 28′ 5″ E), while in AEZ 6, it was carried out in the municipalities of Zè (6°47′00″ N, 2°18′00″ E), Dogbo (6°49′0″ N/1°46′60″ E), Klouékanmè (6°58′49″ N/1°50′32″ E), and Zakpota (7°13′41″ N/2°12′4″ E) (Figure 1).

### 2.2. Sampling of Fall Armyworm Larvae in Maize Fields

Larvae were sampled during the maize growing seasons, from January to February in the dry season and from April to May in the rainy season, in 2021 and 2022. Twelve maize fields were targeted in each AEZ. A total of 384 maize fields at the 2–8 leaf stage (V2–V8) were sampled once over two years during both the dry and rainy seasons, with 96 fields sampled in AEZ 6 and 96 in AEZ.

Maize fields were split into five subplots. Scouting was performed by randomly selecting 6 plants (during the dry season) and 10 plants (during the rainy season) per subplot while moving along a W-shaped pattern [33]. This pattern was used to obtain a more representative assessment of pest incidence and distribution across the entire field. In total, 30 plants and 50 plants were randomly sampled per field during the dry season and rainy seasons, respectively. Inspected maize fields were slightly smaller in size during the dry season (0.5 ha) compared to the rainy season (between 0.5 to 1 ha). The number of larvae and egg masses was recorded per plant. Leaf damage was assessed using a scoring system from 1 to 9, which was first described by [36] and further adjusted by [4] (Table 1). Prior to sampling, farmers were interviewed to identify pesticide-treated maize fields, which were then excluded from the sampling process.

### 2.3. Parameters

Plants with either FAW egg masses, larvae, or visual damage symptoms were considered as infested. The larval infestation rate was calculated for each maize field, then the rate of larval infested fields per AEZ was calculated by considering the total number of fields sampled over each year. The mean percentages of damaged plants per AEZ were determined by considering the sum of percentages of plants showing visible signs of FAW damage per field and the total number of inspected fields in a given year. Plants showing visible signs of FAW damage were categorized as damaged, regardless of the presence of feeding larvae. The mean numbers of larvae per plant per week were determined. Leaf damage severity was also assessed.

### 2.4. Meteorological Data

Monthly mean temperatures and rainfall for the study period in the different study locations were obtained from the Agency for Aerial Navigation Safety (Agence pour la Sécurité de la Navigation Aérienne en Afrique et à Madagascar, ASECNA-Bénin). For temperatures, we used data from the weather station of Bohicon, which is closest to the locations of Ouinhi, Zakpota, Klouékanmè, and Dogbo, while for the localities of Adjohoun, Dangbo, and Zè, we used data from the Cotonou station. For rainfall, we used data from the municipalities of Adjohoun, Dangbo, Ouinhi, and Zakpota, and from the Aplahoué and Allada weather stations for the localities of Dogbo, Klouékanmè, and Zè.

### 2.5. Data Analysis

#### 2.5.1. Larval Dispersion Model Analysis in Maize Fields

The larval dispersion model was analyzed using the Taylor power law and Iwao regression procedure [23,31,37,38,39,40]. Taylor’s power law [40] is expressed as follows:S_i_^2^ = αm_i_β(1)

S_i_^2^ is the variance at a sampling date i; m_i_ is the mean larval density at sampling date i; α is the sampling factor; and β is the aggregation index. The coefficients α and β were estimated by performing linear regression on the log-transformed values [41] of Si^2^ (variance sampling date i), and the corresponding log-transformed values of the mean larval density. The aggregation index, as defined by [42] and used by [31,43], and Ref. [23], was estimated to analyze larval dispersion. The following formula defines this index:m_i_* = m_i_ + [(S_i_^2^/m_i_) − 1],(2)
where m_i_* represents the mean of the aggregation index; m_i_ is the mean larval density per plant for sampling date i; and S_i_^2^ is the variance of larvae at sampling date i. According to [38], many biological distribution models exhibit a linear relationship between m* and m:m* = ϒ + δm,(3)
where ϒ is the regression coefficient; m* is the mean of the aggregation index; and m is the larval density. The slope δ represents the rate of aggregation of the larval population, and ϒ indicates the number of individuals constituting the initial colony [43].

#### 2.5.2. Statistical Analysis

Data on larval infestation rate, and larvae number of AEZ were processed by applying the analysis of variance (ANOVA) based on the general linear model (GLM) with SAS software, version 9.3. Means were separated using the Student–Newman–Keuls (SNK). The t-test was used to compare infestation, severity, and mean number of larvae observed during the two years. The percentage of damaged plants was compared using the least square difference (LSD) test.

## 3. Results

### 3.1. Larval Infestation of Maize by FAW in Dry and Rainy Seasons

The results of the two-factor ANOVA on the maize larval infestation rate by FAW during the dry and rainy seasons are summarized in Table 2. This analysis reveals that larval infestation rates vary significantly according to zone, season, and year (*F_1,317_* = 4.35; *p* < 0.03; *F_1,317_* = 48.20; *p* < 0.0001 and *F_1,317_* = 8.07; *p* < 0.004). Additionally, significant interaction effects were observed between the three factors zone, season, and year (*F_1,317_* = 2.96; *p* < 0.05).

Larval infestation rate displayed a significant difference between the two AEZs for the years 2021 and 2022 (*F_3,129_* = 11.69; *p* < 0.0001). During the dry season, a significantly lower larval infestation rate was observed in 2022 compared to 2021 in each AEZ (Table 3). In the rainy season, the lowest larval infestation rate was observed in AEZ 6 compared to AEZ 8 in 2022 (*F_3,129_* = 3.80; *p* = 0.01) (Table 2). However, no significant difference was obtained between AEZ 6 and 8 in 2021. Larval infestation rates in AEZ 8 were higher during the rainy season compared to the dry season, regardless of year. However, no significant differences were observed between larval infestation rates between AEZ 6 and AEZ 8, regardless of season.

### 3.2. Density of FAW Larvae During the Dry and Rainy Season

The results of the two-factor ANOVA on FAW larval density during the dry and rainy seasons are shown in Table 4. The analysis demonstrates that larval density varies highly significantly with zone, season, and year (*F_1,14752_* = 53.01; *p* < 0.0001; *F_1,14752_* = 235.79; *p* < 0.0001 and *F_1,14752_* = 23.09; *p* < 0.0001). The interactions between zone and season on one side, and between zone, season, and year, were significant when considering larval density (*F_1,14752_* = 66.65; *p* < 0.001 and *F_1,14752_* = 5.59; *p* < 0.0037).

Dry and rainy season FAW larval densities in 2021 and 2022 are presented in Table 5. During the dry season, significant differences were observed between weeks within each AEZ for the same year. Weekly recorded larval densities decreased significantly over time as well, regardless of AEZs. In the rainy season, the larval density of FAW was highest during sampling at weeks 2 and 3 in 2021 in both AEZs and week 4 in AEZs 6 in 2021 (*F*_5,2995_ = 6.38; *p* < 0.001) and 2022 (*F*_9,5391_ = 35.88; *p* < 0.0001) (Table 3). When comparing the overall means, larval density was higher in AZE 8 compared to AZE 6 in the dry season in 2021, but no significant differences were observed between the AEZs in 2022 during the rainy season for larval density. The between-season comparison shows the highest larval densities in AEZ 8 during the dry season while the lowest larval density was recorded during the rainy season, regardless of agroecological zone and year. The analysis of the interactions revealed that larval density was jointly influenced by both zone and season, as well as by zone, season, and year together.

### 3.3. Percentage of Damaged Plants During the Dry and Rainy Seasons

Table 6 presents the results of the two-factor ANOVA on the percentage of damaged plants during the dry and rainy seasons. These results indicate that the percentage of damaged plants varies significantly with season and year (*F_1,317_* =173.08; *p* < 0.0001 and *F_1,317_* = 56.74; *p* < 0001). The interaction between zone, season, and year was significant when considering the percentage of damaged plants (*F_1,317_* = 6.93; *p* < 0.0011).

A comparison of the two AEZs revealed a slight difference in the percentage of damaged plants (*F_3,129_* = 8.09; *p* < 0.0005) (Table 7), regardless of the year.

The percentage of damaged plants was the highest in 2021, regardless of AEZs (*F_3,129_* = 16.61; *p* < 0.0001) (Table 4). However, in 2022, the highest damage was observed in AEZ 8 compared to zone 6. A comparison between seasons revealed more leaf damage during the dry season in 2022, regardless of the agroecological zone.

### 3.4. Leaf Damage Severity During the Dry and Rainy Season

The results of the two-factor ANOVA on leaf damage severity during the dry and rainy seasons are presented in Table 8. Damage severity varies highly significantly with season and year (*F_1,14753_ =* 453.04; *p* < 0.0001 and *F_1,14753_ =* 86.55; *p* < 0.0001). The interaction between zone and season was significant (*F_1,14753_ =* 13.42; *p* < 0.003). Likewise, the interaction between zone, season, and year was significant for damage severity (*F_1,14753_* = 182.58; *p* < 0.0001).

Table 9 shows the severity of FAW damage on maize leaves during the dry and rainy seasons of 2021–2022. During the dry season, in 2021, the highest damage was observed in the third week-sampling in AEZ 8 compared to AEZ 6 (*F_5,1795_ =* 8.60; *p* < 0.0001). But in 2022, even significant differences occurred between weekly samples (*F_9,3231_* = 12.91; *p* < 0.0001), with the highest plant damage obtained in the fourth week in AEZ 6 and from the second to the fifth week in AEZ 8. When comparing leaf damage severity within the same AEZ between seasons, mean values obtained were higher in the dry season compared to the rainy season, regardless of the agroecological zone in 2021 (*F_3,5157_* = 197.99; *p* < 0.0001). In 2022, leaf damage severity was the lowest in AEZ 8 during the rainy season (*F_3,899_* = 13.53, *p* < 0.0001). The interaction between zone and season, as well as between zone, season, and year, demonstrated that leaf damage severity was jointly influenced by zone and season within years.

### 3.5. Dispersion Pattern of FAW Larvae

The linear relationship between the logarithm of S_i_^2^ and the logarithm of m_i_ is depicted in Figure 2 with a determination coefficient (R^2^) of 0.801 (*p* < 0.0001). The slope β of 1.32, exceeding 1, indicates an aggregation trend of larvae. The larval dispersion pattern was found to follow a negative binomial distribution. Figure 3 shows the linear relationship between the aggregation index and the mean larval density per plant, with a determination coefficient (R^2^) of 0.463. The slope δ of the regression line, equal to 1.25, suggests a moderate aggregation of larvae, while the intercept ϒ, equal to 2.08, indicates that FAW larvae were grouped in basic colonies [38,43].

### 3.6. Temperature and Precipitation During the Survey

The dry season had higher monthly mean temperatures than the rainy season, and there were significant variations in rainfall between the two seasons (Figure 4). The monthly mean temperatures at AEZ 6 in 2021 varied from 29.39 to 30.38 °C during the dry season, and 27.48 to 29.84 °C during the rainy season. Monthly mean rainfall varied from 0.25 to 5.12 mm in the dry season, while during the rainy season, it varied from 39.75 to 123.32 mm. For the same area (AEZ 6), temperature variations were observed in 2022, ranging from 26.63 to 29.8 °C in the dry season and from 26.79 to 30.12 °C during the rainy season. Conversely, in the dry season, rainfall varied from 0 to 3.1 mm, whereas during the rainy season, it ranged from 35.95 to 181.62 mm.

Monthly mean temperatures during the dry season in AEZ 8 for 2021 varied from 29.39 to 30.78 °C, while rainy season temperatures ranged from 27.49 to 29.84 °C. Monthly mean rainfall ranged from 4.5 to 5.53 mm during the dry season, while it ranged from 78.06 to 198.03 mm during the rainy season. Temperatures in AEZ 8 ranged from 26.48 to 29.7 °C during the dry season and from 26.79 to 30.12 °C during the rainy season in 2022. Rainfall ranged from 82.73 to 222.9 mm in the wet season and 0 to 0.56 mm in the dry season.

## 4. Discussion

Our study revealed significant variability in FAW larval infestation rates and the percentage of damaged plants across seasons, years, and AEZs, likely influenced by the pest’s life cycle and maize phenology [33,44]. Younger maize growth stages (VE–V3) were particularly susceptible to FAW damage, as evidenced by higher damage scores during early plant development [45,46,47]. Larval infestations were lower in AEZ 6 than in AEZ 8 during the rainy season. Such variability may be due to differences in physical conditions and biotic factors such as food and natural enemies [18].

Interestingly, larval densities during the dry season were much higher than in the rainy season, despite a larger number of maize plants being sampled in the latter. This discrepancy could be attributed to weather factors, including precipitation, temperature, and humidity [18]. Although there was a slight difference in temperatures, substantial differences in rainfall between seasons (Figure 4) likely played a pivotal role in shaping FAW population dynamics. Heavy rainfall may have a lethal effect on larvae, particularly during the 2nd–4th instar stages, as they were frequently observed drowning in the whorls. These findings support the hypothesis that rainfall significantly reduces larval density and plant damage in maize fields, as also reported by [33,48,49]. Seasonal differences in infestation further align with the well-documented impacts of temperature and rainfall on insect life cycles [50,51]. Variations in these climatic factors, exacerbated by climate change, may intensify FAW damage and facilitate its spread to new regions. In Africa, for instance, FAW may exploit migrations from climatically favorable zones to colonize previously unsuitable areas. These findings emphasize the importance of ongoing surveillance and the development of adaptive pest management strategies [52].

Our findings indicate that FAW population density is higher in the dry season than in the rainy season, consistent with studies conducted in other regions of Africa and Latin America. For example, Ref. [33] reported significant seasonal variations in FAW infestations in Uganda and Mozambique, where higher pest densities occurred during the rainy season, mirroring our observations in Southern Benin. Similarly, research from Latin America, where FAW is endemic, demonstrates that rainfall and temperature strongly influence pest dynamics, with warm and humid conditions favoring higher infestation levels [46]. In contrast, a study from Ghana reported an opposite trend, where the rainy season positively influenced FAW populations in maize fields [22]. These contrasting patterns underscore the complexity of pest population dynamics, which are shaped by interacting factors, including pest biology, behavior, and highly variable climatic conditions [18].

Our study further revealed inter-annual variability in larval density and damage severity in AEZ 6 and AEZ 8 during the rainy season. Both larval densities and damage severity were lower in 2022 compared to 2021, likely due to higher rainfall levels recorded in 2022. Heavy rainfall may have drowned many larvae in the whorls, thereby reducing damage to the plants. In contrast, during the dry season, significant variation in FAW larval densities was observed, with lower densities recorded in 2021. This reduction could be attributed to sporadic rains that drowned larvae or the migration of FAW adults driven by unfavorable environmental conditions.

Other important factors that can regulate the FAW population in the field are biotic factors, such as natural enemies, maize varieties, and agricultural practices. These factors significantly shape FAW dynamics [48,53,54].

Larval mortality is influenced by multiple factors, including entomopathogenic fungi, bacterial infections, parasitoids, and predatory insects. For example, *Telenomus remus* Nixon (Hymenoptera; Platygastridae), *Chelonus bifoveolatus* Szépligeti, *Coccygidium luteum* (Brullé), *Cotesia icipe* Fernandez-Triana and Fiaboe (Hymenoptera; Braconidae), *Pristomerus pallidus* (Kriechbaumer), *Charops* sp. (Hymenoptera; Ichneumonidae), and *Drinoquadri zonula* (Thomson) (Diptera; Tachinidae) have been reported as important parasitoids that contribute to the mortality of FAW populations in Benin [55]. Additionally, entomopathogenic fungi such as *Metarhizium* sp., and nematodes such as *Hexamermis* sp., have also been shown to infect FAW larvae [56]. Furthermore, predatory ants have been identified as significant natural enemies, with seven species recorded by [57]. Conservation of natural enemies is crucial for sustainable FAW management [58]. Integrated pest management (IPM) strategies should therefore prioritize practices that enhance the effectiveness of these biocontrol agents.

The population dynamics of FAW in maize fields are strongly influenced by the choice of maize varieties. Conventional varieties are often more susceptible to FAW infestations due to the absence of genetic resistance, leading to higher larval densities and damage levels [59]. In contrast, hybrids and Bt maize varieties exhibit moderate to high resistance, reducing pest survival and oviposition rates [4,60]. For instance, studies have shown that Bt maize containing Cry proteins significantly lowers larval densities by disrupting their feeding [60]. Differences observed in larval densities might have been caused by factors other than variety as the accessibility of resistant varieties remains limited in Benin due to socioeconomic and regulatory challenges, such as the high cost of seeds and restrictions on the adoption of genetically modified crops.

Agricultural practices and cropping systems are factors influencing FAW dynamics, as they directly affect pest habitat and interactions with natural enemies [61]. Fertilizer application can have varying effects on crop damage, depending on the type and quantity of fertilizer used. Inconsistent results may stem from differences in fertilizer application methods. For example, Ref. [62] recently observed that split applications of NPK fertilizers reduced both the incidence and damage caused by FAW. On the other hand, nitrogen-based fertilizers can alter the carbon-to-nitrogen (C/N) ratio in plants, potentially making them more vulnerable to FAW damage [63].

In Benin, smallholder farmers often rotate maize with other crops and intercrop it with cassava and legumes such as cowpea, groundnut, and pigeon pea. Intercropping has been shown to reduce FAW oviposition and feeding activity by disrupting pest habitats and promoting natural enemy populations [64]. Ref. [65] found that intercropping maize with leguminous crops significantly reduced the incidence of both stem borers and FAW. When appropriately designed, diverse cropping systems can effectively maintain FAW populations at manageable levels by combining reduced pest pressure with enhanced natural regulation [66].

Our analysis of the aggregation index and the mean number of larvae per plant revealed a slope of δ = 1.25, indicating a moderate larval aggregation. This aggregation reflects known larval behaviors, such as ballooning during early instars and self-thinning through cannibalism at later stages [67,68]. Environmental factors, including plant morphology, food availability, and microclimatic conditions, also influence larval clustering. These behaviors and environmental interactions align with observations in Mexico [25,27], America [69], and Brazil [26,70]. These findings were similar to previous observations documenting more pronounced intraspecific aggregation patterns in similar pest species [37,71,72]. Furthermore, the intercept of γ = 2.08 supports that FAW larvae evolve in colonies of at least two larvae before they disperse.

Understanding this aggregation pattern is essential for developing effective control strategies, developing effective sampling plans, and predicting pest damage, as this pattern could affect the dispersion of FAW larvae. For instance, probability distribution models based on larval clustering can improve sampling designs and guide site-specific control measures, reducing costs and environmental impacts [73]. Behavioral patterns and environmental characteristics play a key role in determining the spatial distribution of individuals within a population in a given ecosystem. It has been demonstrated that different plant varieties also impact the spatial distribution of insects [74]. Therefore, further investigations are required to validate the spatial distribution of this pest in maize growth stages. The current study did not explore the range of alternative host plants of FAW. Therefore, there is a need to examine the influence of these host plants on the dispersion and population dynamics of this pest insect. The larval density and aggregation patterns of FAW identified in this study can inform action threshold determination for targeted interventions by providing critical data on population dynamics that help identify when pest populations exceed economically damaging levels, allowing for timely and precise management strategies.

The current study provides valuable insights into the seasonal dynamics and key factors influencing FAW infestations in maize crops across two AEZs in Benin. The findings confirm that FAW is present year-round, with larval populations peaking during the rainy season and declining during the dry season, likely due to temperature and rainfall variations. These results underscore the need for tailored management strategies that account for seasonal patterns. Biological control could be optimized by employing natural enemies and entomopathogens adapted to rainy season conditions, reducing FAW’s impact while minimizing reliance on chemical inputs. Early detection and control measures during the rainy season are critical, particularly in zones with consistently high infestation rates. This study provides valuable baseline data on the spatiotemporal dynamics of FAW in southern Benin, contributing to a better understanding of its behavior and distribution patterns. The findings lay the groundwork for further research on integrated pest management strategies, particularly in similar agroecological zones in sub-Saharan Africa and beyond.

## Figures and Tables

**Figure 1 insects-16-00145-f001:**
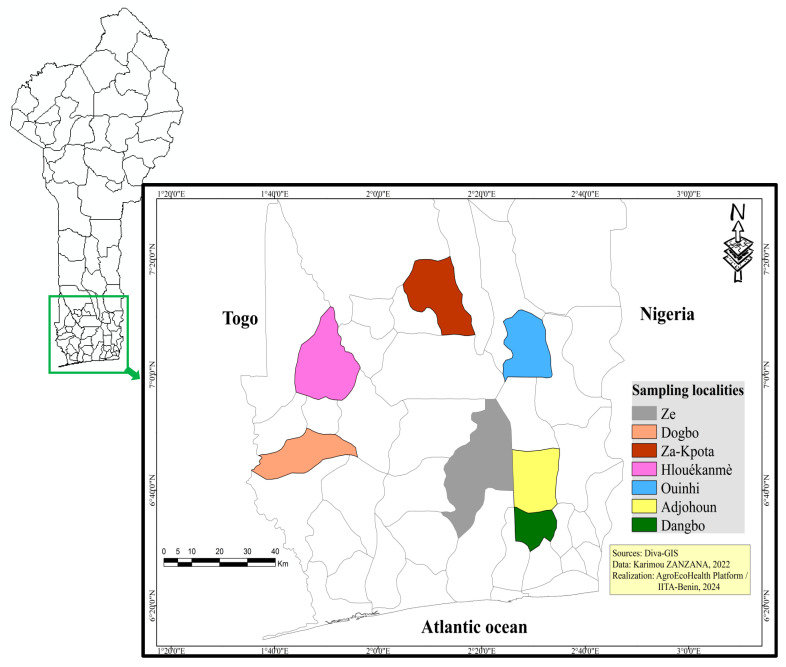
Study municipalities in southern Benin.

**Figure 2 insects-16-00145-f002:**
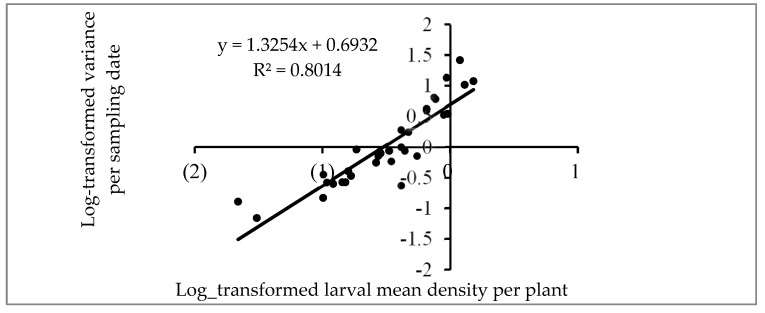
Relationship between the log-transformed values of Si^2^, variance for the plant at each sampling date, and the corresponding log-transformed values (m_i_) of larval mean density per plant.

**Figure 3 insects-16-00145-f003:**
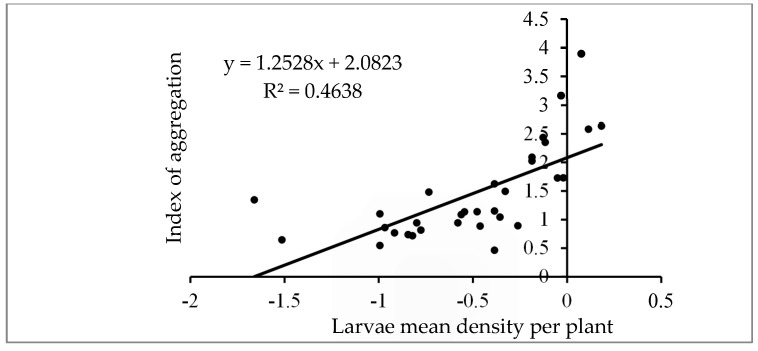
Relationship between the index of aggregation (m_i_*) and the larval mean density per plant (m_i_).

**Figure 4 insects-16-00145-f004:**
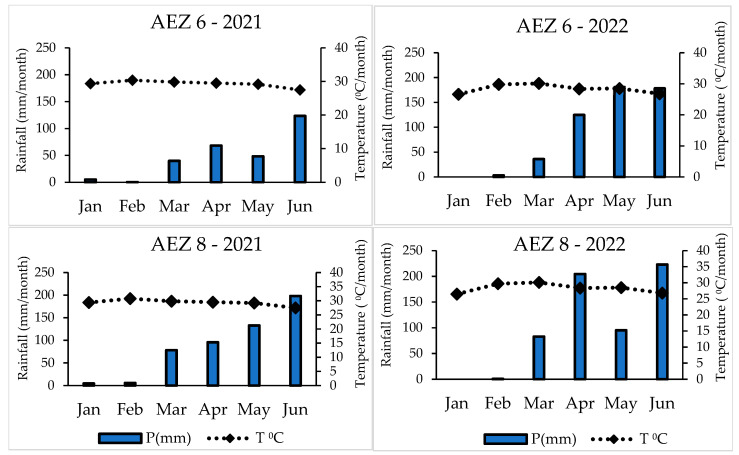
Monthly mean temperatures (°C) and mean rainfall (mm) in the AEZ in the dry season (January to February) and in the rainy season (March to June).

**Table 1 insects-16-00145-t001:** Scale for assessment of leaf damage due to FAW in maize [4] adjusted from [36].

Score	Description
1	No visible leaf-feeding damage
2	Few pinholes on 1–2 older leaves
3	Several shot-hole injuries on a few leaves (<5 leaves) and small circular hole damage to leaves
4	Several shot-hole injuries on several leaves (6–8 leaves) or small lesions/pinholes, small circular lesions, and a few small elongated (rectangular-shaped) lesions up to 1.3 cm in length on whorl and furl leaves
5	Elongated lesions (>2.5 cm long) on 8–10 leaves, plus a few small- to mid-sized uniform to irregular-shaped holes (with basement membrane consumed) eaten from the whorl and/or furl leaves
6	Several large, elongated lesions on several whorl and furl leaves and/or several large uniform- to irregular-shaped holes eaten from furl and whorl leaves
7	Many elongated lesions of all sizes on several whorl and furl leaves plus several large uniform- to irregular-shaped holes eaten from the whorl and furl leaves
8	Many elongated lesions of all sizes on most whorl and furl leaves plus many mid- to large-sized uniform- to irregular-shaped holes eaten from the whorl and furl leaves
9	Whorl and furl leaves almost totally destroyed, with the plant dying as a result of extensive foliar damage

**Table 2 insects-16-00145-t002:** Analysis of variance (Fisher’s value) results for larval infestation of maize by FAW in dry and rainy seasons.

Source	DF	Mean Square	*F* Value	*p*
Zone	1	0.15	4.35	0.0378
Replicate	59	0.04	1.21	0.1546
Season	1	1.67	48.20	0.0001
Year	1	0.28	8.07	0.0048
Zone*Season	1	0.11	3.36	0.0677
Zone*Year	1	0.00	0.19	0.6637
Zone*Season*Year	2	0.10	2.96	0.0531

**Table 3 insects-16-00145-t003:** Larval infestation rate in the AEZ 6 and 8 during the dry and rainy seasons.

Season	Agroecological Zone	Larval Infestation Rate (%) (Mean ± SE)		
		Year 2021	Year 2022	
Dry season	AEZ 6	36.94 ± 2.61 bB	29.97 ± 2.25 aC	*F_3,129_* = 11.69; *p* < 0.0001
	AEZ 8	36.50 ± 2.00 bB	29.87 ± 2.27 aC	
Rainy season	AEZ 6	42.40 ± 2.90 aB	41.44 ± 2.32 aB	*F_3,129_* = 3.80; *p* < 0.01
	AEZ 8	47.96 ± 2.87 abA	50.94 ± 1.90 bA	
		*F_3,10_* = 3.54; *p* < 0.01	*F_3,17_* = 21.49; *p* < 0.0001	

AEZ = agroecological zone; SE = standard error. Means ± SE followed by the same uppercase letter in the column are not significantly different. Mean ±SE followed by the same letter are not significantly different at the 5% threshold (*p* < 0.05). Uppercase letters refer to the comparison between seasons within the same year while lowercase letters refer to the comparison of agroecological zones and years for the same season.

**Table 4 insects-16-00145-t004:** Analysis of variance (Fisher’s value) results for the density of FAW larvae during the dry and rainy seasons.

Source	DF	Mean Square	*F* Value	*p*
Zone	1	60.77	53.01	0.0001
Replicate	599	1.11	0.97	0.6681
Season	1	270.33	235.79	0.0001
Year	1	26.47	23.09	0.0001
Zone*Season	1	76.41	66.65	0.0001
Zone*Year	1	2.11	1.84	0.1744
Zone*Season*Year	2	6.41	5.59	0.0037

**Table 5 insects-16-00145-t005:** Weekly larval density of FAW in the two AEZs during the dry and rainy seasons.

Season	Agroecological Zone	Sampling Week	Larval Density (Mean ± SE)	
			Year 2021	Year 2022
Dry season	AEZ 6	1	0.88 ± 0.08 abc	0.96 ± 0.09 abc
		2	0.58 ± 0.05 d	0.68 ± 0.05 cde
		3	0.71 ± 0.06 cd	0.67 ± 0.06 cde
		4	-	0.56 ± 0.05 ed
		5	-	0.43 ± 0.04 e
	Overall mean		0.73 ± 0.03 B	0.66 ± 0.02 B
	AEZ 8	1	1.11 ± 0.09 a	1.07 ± 0.11 ab
		2	0.82 ± 0.07 bc	1.20 ± 0.09 a
		3	0.98 ± 0.70 ab	0.95 ± 0.07 abc
		4	-	0.87 ± 0.08 bc
		5	-	0.82 ± 0.07 bcd
	Overall mean		0.98 ± 0.04 A	0.99 ± 0.03 A
			*F_5,1795_* = 7.15; *p* < 0.0001	*F_9,3230_* = 10.42; *p* < 0.0001
Rainy season	AEZ 6	1	0.44 ± 0.03 b	0.18 ± 0.03 d
		2	0.62 ± 0.04 a	0.42 ± 0.03 c
		3	0.69 ± 0.04 a	0.36 ± 0.03 c
		4	-	0.38 ± 0.03 c
		5	-	0.77 ± 0.04 a
	Overall mean		0.59 ± 0.02 C	0.43 ± 0.01 C
	AEZ 8	1	0.48 ± 0.04 b	0.37 ± 0.03 c
		2	0.56 ± 0.04 a	0.21 ± 0.02 d
		3	0.63 ± 0.04 a	0.39 ± 0.03 c
		4	-	0.45 ± 0.03 c
		5	-	0.67 ± 0.03 b
	Overall mean		0.56 ± 0.02 C	0.42 ± 0.01 C
			*F_5,2995_* = 6.38; *p* < 0.0001	*F_9,5391_* = 35.88; *p* < 0.0001

AEZ = agroecological zone; SE = standard error. Means ± SE followed by the same uppercase letter in the column are not significantly different. Mean ± SE followed by the same letter in the column is not significantly different at the 5% threshold (*p* < 0.05). Uppercase letters refer to the comparison of the overall mean between seasons within the same year and lowercase letters refer to the comparison of agroecological zones within the same year regardless of the season.

**Table 6 insects-16-00145-t006:** Analysis of variance (Fisher’s value) results for the percentage of damaged plants during the dry and rainy seasons.

Source	DF	Mean Square	*F* Value	*p*
Zone	1	0.07	2.71	0.1010
Replicate	59	0.03	1.16	0.2095
Season	1	4.68	173.08	0.0001
Year	1	1.53	56.74	0.0001
Zone*Season	1	0.08	3.05	0.0819
Zone*Year	1	0.09	3.36	0.0679
Zone*Season*Year	2	0.18	6.93	0.0011

**Table 7 insects-16-00145-t007:** Percentage of damaged plants by FAW in the two AEZs during the dry and rainy seasons.

Season	Agroecological Zones	Percentage of Damaged Plants (Mean ± SE)		
		Year 2021	Year 2022	
Dry season	AEZ 6	58.33 ± 1.51 bB	51.53 ± 2.31 aC	*F_3,129_* = 8.09; *p* < 0.0005
	AEZ 8	55.72 ± 1.82 abB	53.73 ± 2.35 aC	
Rainy season	AEZ 6	80.92 ± 1.94 cA	63.44 ± 2.06 aB	*F_3,129_* = 16.61; *p* < 0.0001
	AEZ 8	82.22 ± 2.36 cA	72.05 ± 1.86 bA	
		*F_3,10_* = 66.42; *p* < 0.0001	*F_3,17_* = 19.91; *p* < 0.0001	

AEZ = agroecological zone; SE = standard error. Means ± (SE) followed by the same uppercase letter in the column are not significantly different. SE = standard error. Mean ± (SE) followed by the same letter in the column are not significantly different at the 5% threshold (*p* < 0.05). Uppercase letters refer to the comparison of the overall mean between seasons within the same year and lowercase letters refer to the comparison of agroecological zones within the same year regardless of the season.

**Table 8 insects-16-00145-t008:** Analysis of variance (Fisher’s value) results in leaf damage severity during the dry and rainy seasons.

Source	DF	Mean Square	*F* Value	*p*
Zone	1	4.06	1.71	0.1913
Replicate	599	2.49	1.05	0.1981
Season	1	1077.71	453.04	0.0001
Year	1	205.88	86.55	0.0001
Zone*Season	1	31.91	13.42	0.0003
Zone*Year	1	2.86	1.20	0.2724
Zone*Season*Year	2	434.31	182.58	0.0001

**Table 9 insects-16-00145-t009:** Weekly leaf damage severity in the two AEZs during the dry and rainy seasons.

Season	Agroecological Zone	Sampling Week	Plant Damage Score Per AEZ (Scale 1–9) (Mean ± SE)	
			Year 2021	Year 2022
Dry season	AEZ 6	1	3.36 ± 0.10 b	2.58 ± 0.07 b
		2	3.17 ± 0.09 bc	2.25 ± 0.06 c
		3	3.30 ± 0.08 bc	2.30 ± 0.06 c
		4	-	2.73 ± 0.07 ab
		5	-	2.24 ± 0.07 c
	Overall mean		3.28 ± 0.05 A	2.42 ± 0.03 BC
	AEZ 8	1	2.95 ± 0.08 c	2.32 ± 0.07 c
		2	3.13 ± 0.08 bc	2.63 ± 0.07 ab
		3	3.82 ± 0.15 a	2.63 ± 0.06 ab
		4	-	2.89 ± 0.07 a
		5	-	2.84 ± 0.07 a
	Overall mean		3.30 ± 0.06 A	2.66 ± 0.03 A
			*F_5,1795_* = 8.60; *p* < 0.0001	*F_9,3231_* = 12.91; *p* < 0.0001
Rainy season	AEZ 6	1	2.03 ± 0.05 b	1.55 ± 0.04 f
		2	2.10 ± 0.05 b	2.56 ± 0.07 b
		3	2.32 ± 0.05 a	2.70 ± 0.08 b
		4	-	1.98 ± 0.05 e
		5	-	3.48 ± 0.08 a
	Overall mean		2.16 ± 0.02 B	2.46 ± 0.03 B
	AEZ 8	1	2.00 ± 0.05 b	1.84 ± 0.04 e
		2	2.10 ± 0.05 b	1.79 ± 0.05 e
		3	2.32 ± 0.05 a	2.45 ± 0.07 c
		4	-	2.23 ± 0.05 d
		5	-	3.41 ± 0.08 a
	Overall mean		2.14 ± 0.03 B	2.35 ± 0.02 C
			*F_5,2995_* = 8.22; *p* < 0.0001	*F_9,5391_* = 110.90; *p* < 0.0001

AEZ = agroecological zone; SE = standard error. Means ± (SE) followed by the same uppercase letter in the column are not significantly different. Mean ± (SE) followed by the same letter in the column are not significantly different at the 5% threshold (*p* < 0.05). Uppercase letters refer to the comparison of the overall mean between seasons within the same year and lowercase letters refer to the comparison of agroecological zones within the same year regardless of the season.

## Data Availability

The original contributions presented in this study are included in the article. Further inquiries can be directed to the corresponding authors.

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
