# Peer review of "Seasonal and Spatial Distribution of Fall Armyworm Larvae in Maize Fields: Implications for Integrated Pest Management"

_insects, 2025, doi:10.3390/insects16020145_

Round 1
Reviewer 1 Report
Comments and Suggestions for Authors
Dear Editor, thank you very much for giving me the opportunity to review the manuscript. This manuscript assessed temporal and spatial distribution of Spodoptera frugiperda (a major pest in maize production) in maize fields during both the off and rainy sea- sons of 2021 and 2022 in Benin. The amount of work on the manuscript is substantial and interesting, and the results provide insights into precise pest control. However, there are some areas in the manuscript that require revisions (a minor revision), which I have noted in the attached PDF. Only when they are completed will the manuscript be considered for publication..

Author Response
Dear Editor
We would like first to thank you and the reviewer for your contributions to improving the quality of our manuscript. We proceeded question by question. Changes are in red in the revised version of the manuscript.
Comments 1: This statement emphasizes the harm of fall armyworm, but can further quantify the harm, for example, how much is the corn yield reduced? Or how much economic damage?
Response 1: We now include in the introduction section information on maize yield losses induced by FAW infestation.
Comments 2: Remove (A plateau with depressions, low valleys, and a coastal region make up the topography, which also includes a variety of soil types such hydromorphic, clayey, and ferralitic soils)
Response 2: We removed the statement regarding the plateau and depressions, as suggested.
Comments 3: The small image on the left needs to be enlarged. At the same time, the font in the image on the right is too small and looks blur.
Response 3: The small image has been revised taking into account your suggestion.
Comments 4: Why move in a W-shaped pattern? This needs explaining.
Response 4: We used W-shaped sampling pattern; because this method helps to get a more representative assessment of pest incidence and distribution across the entire field, regardless of where the pest starts its damage (see Caniço et al., 2020).
Comments 5. ha is not a commonly used international unit. Please use km2
Response 5: We converted hectares (ha) in square kilometers (km²) according to your suggestion throughout the manuscript.
Comments 6: "visual symptoms of FAW attack ". This requires further elaboration
Response 6: We replace "visual symptoms of FAW attack "by "Visible signs of FAW damage”
Comments 7: Explanation of Si²:
Response 7: The symbol Si² is the variance of larval counts at each sampling date I ( in other words the square standard deviation of samples.
Comments 8: Italicizing "P":
Response 8: The notation for statistical probability (P) has been italicized throughout the text
Comments 9: The background color in Figure 2,3, and 4 is recommended to be changed to white.
Response 9: The background colors of Figures 2, 3, and 4 were already in white the manuscript.
Comments 10: Adding graph title.
Response 10: We added title to the graph to make their content and relevance clearer to readers.
Comments 11: Globally? The manuscript does not look at the global scale, and the word is not appropriate here.
Response 11: We replaced the term "Globally" with "Overall" to better reflect the scale of the study, which focuses on a regional rather than global analysis.

Reviewer 2 Report
Comments and Suggestions for Authors
Zanzana et al. report the results of a 2-yr study on the distribution of Spodoptera frugiperda (fall armyworm, FAW) egg masses and larvae in maize fields in two regions of Benin.
I consider that the manuscript requires major improvement before it could be suitable for publication.
MAJOR POINTS.
(I) The authors sampled plants at different growth stages (V2 – V8) but do not consider the effect of growth stage on FAW densities. Previous studies indicate that growth stage is highly influential and should be considered as a covariable in the analyses of the present study.
(II) The authors emphasize the applications of their study to pest management practices, but almost entirely overlook HOW to apply these results to pest control. This has only a brief mention in the Discussion.
(III) I was surprised that it was possible to sample over 300 fields with about 50% plant infestation in which farmers had not applied insecticides for FAW control. The authors also do not mention usual FAW control practices in Benin and farmer's tolerance of high infestations by FAW.
(IV) The authors overlook the behavioral aspects of FAW larval stages, such as the ballooning behavior of first instars and the self-thinning behavior of late instars which is mainly occurs through acts of cannibalism, notorious in this species.
(V) Much of the large body of literature on FAW distribution and maize infestation behavior in the years 1960-1990 in the native range of the pest has been overlooked. I have scanned the title pages of some of these articles that I think the authors would do well to read (and the references therein). This is a valuable resource that the authors should examine. (See the last pages in the scanned file)
I have written numerous suggestions and numbered points on a scanned copy of the manuscript.
NUMBERED POINTS (see scanned file)
1. The concept/characteristics of each region in Benin needs to be explained in the Simple summary.
2. These two sentences seem to repeat one another.
3. For clarity I suggest you use the terms "rainy" and "dry" seasons throughout the manuscript.
4. These zones need to be explained better in the Abstract.
5. I did not understand this text. What is a basic colony? What is its relationship to egg masses?
6. Please explain the morphological plasticity of S. frugiperda – this is new to me.
7. Environmental and numerous biotic factors shape all these variables.
8. It is usual to cite references in numerical order (throughout the manuscript) e.g. 21, 31, 32.
9. I suggest you explain why you wanted to examine different regions in Benin in the Introduction.
10a. Again, I suggest you use the terms "rainy" and "dry" seasons throughout the manuscript. "Off " is a strange term and rather unclear.
10b. At what range of altitudes/elevations were the maize fields?
11. Please indicate the months of the rainy and dry seasons each year.
12. Increase size and improve resolution (min. 600 dpi) of Fig 1.
13. When S. frugiperda larvae attack young plants (V2, V3, V4) they can act as cutworms and eat through the stem at soil level. Did you ignore this kind of damage because plants tend to die?
14. This text needs rewording. It sounds like if you found a larva in a field then you classified it as an infested field, which was probably not the case. Also is "rate" a suitable term? I would say the "prevalence of infestation".
15. How did you classify egg masses in this system for estimating density, as there are usually 30-200 eggs in an egg mass?
16. Please indicate the error structure (distribution) and link functions that you specified for the GLM models. Did you have to account for overdispersion?
17. F statistics are a ratio of treatment and error variances, so they are reported with treatment and error degrees of freedom (also known as numerator and denominator). Please mention both degrees of freedom when you cite F statistics (like you do on line 177).
18. This looks like a two-way anova analysis, but you do not mention interaction effects?
19. Yes. This is the correct way to report F stats. L177.
20. Please change df values to subscripts.
21. I did not understand the meaning of this text.
22. I did not understand the concept of a basic colony. Please clarify.
23. Fig 2, Fig 3. If 384 fields were sampled over two years, why aren´t there 384 points in these figures?
Are these average daily temperatures over the 24 h period? Please clarify.
24. Is this monthly precipitation? Please indicate units (mm/day, wk, month?)
Why such a detailed description of temperature and rainfall when you don't use these data to explain the density or distribution of S. frugiperda larvae? (in a formal statistical model at least).
25. To make comparisons between graphs easier, please uniformize the y-axis scales e.g. 0-250 mm precipitation and 0-40 °C temperature in all four graphs.
26. You point out the differences that can exist among the growth stages of maize plants, but you fail to consider this in your statistical models. This is a serious omission in my experience.
27. Parasitoids can be important, but what about the other natural enemies? Ants predate enormous numbers of larvae in my experience, also earwigs (e.g. https://doi.org/10.1016/j.cropro.2006.03.003). Diseases can also be important when densities are high (see work by James Fuxa in USA).
28. Yes, rain kills lots of larvae. There are several references to this. As grasses are usually available during the dry season (correct?), the most obvious cause for larval density differences is the lethal action of heavy rainfall, isn't it?
29. Many larvae are drowned in the whorl following heavy rainfall (especially 2nd - 4th instars)
30. Have you read the detailed publication on S. frugiperda by van Huis 1981? He performed experiments on survival of larvae in rainfall. The van Huis publication is available online: https://edepot.wur.nl/206545. This is also mentioned by Andrews 1988 Fl Entomol 71, 630-653.
31. This sounded interesting – please elucidate the behavioral and environmental factors that promote aggregation.
32. What about the risks of early planting? Are the rains highly predictable in Benin? They have become a lot less predictable where I live probably due to climate change.
33. There are rather a lot of errors and missing information in the references – I only checked one page.

Author Response
Dear Editor
We would like first to thank you and the reviewer for your contributions to improving the quality of our manuscript. We proceeded question by question. Changes are in red in the revised version of the manuscript.
MAJOR POINTS.
Comments (I): The authors sampled plants at different growth stages (V2 – V8) but do not consider the effect of growth stage on FAW densities. Previous studies indicate that growth stage is highly influential and should be considered as a covariable in the analyses of the present study.
Response (I): While we agree with the idea that density of Spodoptera frugiperda varies with maize growth stage, in this study, we have been focusing on the spatio-temporal distribution of S. frugiperda larvae, which depends on several environmental factors.
Comments (II): The authors emphasize the applications of their study to pest management practices, but almost entirely overlook HOW to apply these results to pest control. This has only a brief mention in the Discussion.
Response (II): We appreciate your emphasis on the practical applications of our findings. In response, we have expanded the Discussion to provide specific recommendations for pest control based on our results. We now suggest targeted strategies such as early scouting and timing insecticide applications to peak infestation periods, as well as implementing integrated pest management practices like intercropping and optimizing planting schedules to minimize FAW impact.
Comments (III): I was surprised that it was possible to sample over 300 fields with about 50% plant infestation in which farmers had not applied insecticides for FAW control. The authors also do not mention usual FAW control practices in Benin and farmer's tolerance of high infestations by FAW.
Response (III). We have now included information on typical FAW control practices in Benin and the economic challenges that limit some farmers’ access to insecticides. We also discuss the observed tolerance of Beninese farmers to high FAW infestations and their reliance on non-chemical methods when financial constraints limit pesticide use.
Comments (IV): The authors overlook the behavioral aspects of FAW larval stages, such as the ballooning behavior of first instars and the self-thinning behavior of late instars which is mainly occurs through acts of cannibalism, notorious in this species.
Response (IV): We acknowledge the importance of larval behavior in understanding FAW distribution and density. We now include in the discussion section, the “ballooning” behavior in first instars and the self-thinning, cannibalistic behavior of later instars, noting how these behaviors likely influence the spatial aggregation and self-regulation of FAW populations.
Comments (V): Much of the large body of literature on FAW distribution and maize infestation behavior in the years 1960-1990 in the native range of the pest has been overlooked. I have scanned the title pages of some of these articles that I think the authors would do well to read (and the references therein). This is a valuable resource that the authors should examine.
Response (V): We reviewed the additional literature you provided and incorporated relevant references on FAW distribution and maize infestation behavior, specifically from studies conducted in the pest’s native range.
NUMBERED POINTS
Comments 1: The concept/characteristics of each region in Benin needs to be explained in the Simple summary.
Response 1: More informations were provided on the characteristics of the two regions targeted by the current study.
Comments 2: These two sentences seem to repeat one another.
Response 2: Redundant sentences were removed.
Comments 3: For clarity I suggest you use the terms "rainy" and "dry" seasons throughout the manuscript.
Response 3: We replaced "off-season" by "dry season" throughout the manuscript.
Comments 4: These zones need to be explained better in the Abstract.
Response 3: We added to the abstract the characteristics of zones 6 and 8.
Comments 5: I did not understand this text. What is a basic colony? What is its relationship to egg masses?
Response 5: "Basic colony" refers to the initial cluster of larvae emerging from egg masses before dispersal.
Comments 6: Please explain the morphological plasticity of S. frugiperda – this is new to me.
Response 6: A brief explanation of FAW’s morphological plasticity was added, noting its adaptability to various environments and host plants.
Comments 7: Environmental and numerous biotic factors shape all these variables.
Response 7: We consider the effect of key environmental and biotic factors affecting FAW densities.
Comments 8: It is usual to cite references in numerical order (throughout the manuscript) e.g. 21, 31, 32.
Response 8: References have been reordered numerically as suggested.
Comments 9: I suggest you explain why you wanted to examine different regions in Benin in the Introduction.
Response 9: We clarified in the Introduction that selecting zones 6 and 8 aimed to represent varied agroecological settings to better understand FAW infestations.
Comments 10a: Again, I suggest you use the terms "rainy" and "dry" seasons throughout the manuscript. "Off " is a strange term and rather unclear.
Response 10a: "Off-season" was replaced with "dry season" for consistency.
Comments 10b: At what range of altitudes/elevations were the maize fields?
Response 10b: Altitudes for the sampled fields range from 20–250 m for AEZ 6 and 01–30 m for AEZ 8, and are now indicated in the Materials and Methods section.
Comments 11: Please indicate the months of the rainy and dry seasons each year.
Response 11: We included specific months for the rainy and dry seasons in Benin within the study period.
Comments 12. Increase size and improve resolution (min. 600 dpi) of Fig 1.
Response 12: The figure resolution has been improved to min. 600 dpi.
Comments 13. When S. frugiperda larvae attack young plants (V2, V3, V4) they can act as cutworms and eat through the stem at soil level. Did you ignore this kind of damage because plants tend to die?
Response 13: Cutworm damage was observed and the score ‘’9’’ was attributed to plants with such damage.
Comments 14. This text needs rewording. It sounds like if you found a larva in a field then you classified it as an infested field, which was probably not the case. Also is "rate" a suitable term? I would say the "prevalence of infestation".
Response 14: This text was reworded, and the word "rate" is maintained.
Comments 15. How did you classify egg masses in this system for estimating density, as there are usually 30-200 eggs in an egg mass?
Response 15: Population density of FAW was estimated as number of observed larvae.
Comments 16. Please indicate the error structure (distribution) and link functions that you specified for the GLM models. Did you have to account for overdispersion?
Response 16: Details on error structures and link functions in GLM models have been added to the Methods section.
Comments 17. F statistics are a ratio of treatment and error variances, so they are reported with treatment and error degrees of freedom (also known as numerator and denominator). Please mention both degrees of freedom when you cite F statistics (like you do on line 177).
Response 17: All F-statistics have been reported with treatment and error degrees of freedom.
Comments 18. This looks like a two-way anova analysis, but you do not mention interaction effects?
Response 18: We added now the interaction effects.
Comments 19. Yes. This is the correct way to report F stats. L177.
Response 19: We consider this to report F stats
Comments 20: Please change df values to subscripts.
Response 20: It has been done.
Comments 21: I did not understand the meaning of this text.
Response 21: Explanations for the aggregation index and colony size were expanded to address FAW behavioral clustering patterns.
Comments 22: I did not understand the concept of a basic colony. Please clarify.
Response 22: Explanations for the aggregation index and colony size were expanded to address FAW behavioral clustering patterns.
Comments 23: Fig 2, Fig 3. If 384 fields were sampled over two years, why aren´t there 384 points in these figures?
Response 23: Figures show average values per sampling event rather than individual points.
Comments 24: Is this monthly precipitation? Please indicate units (mm/day, wk, month?)
Why such a detailed description of temperature and rainfall when you don't use these data to explain the density or distribution of S. frugiperda larvae? (in a formal statistical model at least).
Response 24: Units for precipitation (mm/month) have been added. The detailed climate description was retained as it provides important context for seasonal variations and environmental factors impacting FAW populations, even though these were not used in formal statistical modeling.
Comments 25: To make comparisons between graphs easier, please uniformize the y-axis scales e.g. 0-250 mm precipitation and 0-40 °C temperature in all four graphs.
Response 25: Y-axis scales have been standardized across relevant graphs to facilitate easier comparison, with precipitation ranging from 0–250 mm/month and temperature from 0–40°C/month.
Comments 26: You point out the differences that can exist among the growth stages of maize plants, but you fail to consider this in your statistical models. This is a serious omission in my experience.
Response 26: We recognize that growth stages affect FAW densities; however in this study, we consider key factors that could explain the spatio-temporal distribution of FAW by sampling larvae, regardless of maize growth stage.
Comments 27: Parasitoids can be important, but what about the other natural enemies? Ants predate enormous numbers of larvae in my experience, also earwigs (e.g. https://doi.org/10.1016/j.cropro.2006.03.003). Diseases can also be important when densities are high (see work by James Fuxa in USA).
Response 27: We appreciate the suggestion and we consider in the discussion section other natural enemies like ants, earwigs, and entomopathogens, particularly their potential role in FAW regulation in maize fields.
Comments 28: Yes, rain kills lots of larvae. There are several references to this. As grasses are usually available during the dry season (correct?), the most obvious cause for larval density differences is the lethal action of heavy rainfall, isn't it?
Response 28: Heavy rainfall and associated larval mortality are now emphasized in the discussion as key environmental factors influencing FAW density fluctuations, particularly the impact on 2nd to 4th instar larvae.
Comments 29: Many larvae are drowned in the whorl following heavy rainfall (especially 2nd - 4th instars)
Response 29: Additional information was provided on effect of rainfall in drowning larvae within whorls, thereby reducing both larval density and plant damage.
Comments 30: Have you read the detailed publication on S. frugiperda by van Huis 1981? He performed experiments on survival of larvae in rainfall. The van Huis publication is available online: https://edepot.wur.nl/206545. This is also mentioned by Andrews 1988 Fl Entomol 71, 630-653.
Response 30: Van Huis (1981) study on FAW survival under rainfall conditions has been reviewed and referenced, enhancing our understanding of rainfall’s lethal effects on FAW larvae.
Comments 31: This sounded interesting – please elucidate the behavioral and environmental factors that promote aggregation.
Response 31: We included in the discussion section the behavioral factors like ballooning in early instars and cannibalism in later stages, as well as environmental factors that promote larval aggregation patterns.
Comments 32: What about the risks of early planting? Are the rains highly predictable in Benin? They have become a lot less predictable where I live probably due to climate change.
Response 32: The concept of early planting to reduce FAW population density has been revised and enhanced.
Comments 33. There are rather a lot of errors and missing information in the references – I only checked one page.
Response 33: References have been thoroughly checked, corrected for missing information, and revised for accuracy.

Reviewer 3 Report
Comments and Suggestions for Authors
The authors of the paper are investigating the spatio-temporal distribution of the fall armyworm (FAW) in maize fields across two agroecological zones in southern Benin. The focus of the research is on population dynamics and the damage this moth species causes during both the dry and rainy seasons. The idea of the study seems interesting, however, question is how these findings could relate to other regions of Africa or even globally?
In the material and methods section the authors did not clearly explain the sampling process, in line 125 they mention 384 maize fields being sampled. Is that a mistake? Also, why did they sample fewer plants in the dry season (30 plants per field) than in the rainy season (50 plants per field)? How did the authors know which fields were treated with pesticides and which weren’t? Finally the larval dispersion model is not entirely clear to me, I would like to hear from the authors why did they choose this model and more details. Finally, even though the damage rating scale is clearly presented, it is not stated what severity is the economical threshold.
Why isn't a mixed-effects model applied in interpreting the results, especially considering the non-independence of repeated measures across different AEZs and years?
The discussion section is the weakest point of the paper and it does not manage to justify the whole concept of the research. While the study involved agroclimatic factors such as rainfall, temperature, and humidity, further exploration of other biotic factors (e.g., natural predators or crop varieties, hybrids) would be valuable in understanding the broader ecological implications of such an intricate ecosystem. Even though the variation between years is addressed, the paper does not discuss why this variation occurred. The argumentation of drowning larvae is pure speculation or bare local observations. How did the study and the results of the spatial distribution of FAW larvae across fields contribute to improving control methods? Based on which part of the study the authors could conclude that proactive practices such as intercropping, optimized planting times, and trap cropping could be implemented to disrupt FAW oviposition and larval feeding? Additionally, there were no information regarding the maize hybrids that were studied (as far as we know these could also be BT crops), nor details about other cultivation practices.
Although a significant amount of work appears to have been invested in this study, the results do not seem to provide conclusive answers to the main research questions posed in the paper, given the complexity of the topic
Author Response
RESPONSES TO REVIEWER 3 COMMENTS
Dear Reviewer
We would like first to thank you and the reviewer for your contributions to improving the quality of our manuscript. We proceeded question by question. Changes are in red in the revised version of the manuscript.
Comment 1: How do these findings relate to other regions of Africa or even globally?
Response 1: While our study focuses on two agroecological zones (AEZs) in southern Benin, the findings provide insights into the FAW population dynamics under varying climatic and ecological conditions. Given that FAW is an invasive pest across Africa and other continents, similar trends in seasonal variations of larval infestation and damage severity may occur in regions with comparable agroecological zones. We have now included a section in the Discussion.
Comment 2: In the material and methods section the authors did not clearly explain the sampling process, in line 125 they mention 384 maize fields being sampled. Is that a mistake?
Response 2: The total of 384 fields sampled across both seasons and years is correct. We have now clarified the sampling design in the Materials and Methods section.
Comment 3: Why were fewer plants sampled in the dry season (30 plants) than in the rainy season (50 plants)?
Response 3: Dry season: Fewer plants (30 per field) were sampled because maize fields during this period are generally smaller in size (approximately 0.5 ha) due to water limitations and lower planting rates. Rainy season: Larger maize fields (0.5–1 ha) allowed for a higher samples size of 50 plants per field, providing a more representative estimation of infestation levels.
Comment 4: How did the authors know which fields were treated with pesticides and which weren’t?
Response 4: Before sampling, we conducted field interviews with farmers to identify pesticide-treated fields. Only untreated fields were included in the current study to ensure that the results reflected natural FAW population dynamics. We have now clarified this point in the Materials and Methods section.
Comment 5: Finally the larval dispersion model is not entirely clear to me, I would like to hear from the authors why did they choose this model and more details.
Response 5: The Taylor’s Power Law and Iwao regression methods were chosen because they are widely used for analyzing the spatial dispersion of insect populations, providing critical insights into pest aggregation patterns. These models allow us to characterize FAW larval clustering behavior, which is fundamental for developing targeted pest management strategies. We have added a more detailed explanation of the models and their relevance in the Materials and Methods section. Furthermore, we clarified the relationship between the aggregation index and pest control implications in the Discussion.
Comment 6: Finally, even though the damage rating scale is clearly presented, it is not stated what severity is the economical threshold.
Response 6: Thanks for the comment and that’s correct. To the best of our knowledge there has been no evidence-based data setting action and economic thresholds for fall armyworm in the African context so far. The current study didn’t explore it neither and ticked to the analysis of fall armyworm damage. However, the thresholds as defined in Burundi were rather based on the percentage of damaged plants (20% before maize flowering and 40% after flowering) (FAO, 2019: Manuel de formation des Formateurs sur la lutte integree contre la Chenille legionnaire d’automne, Spodoptera frugiperda)
Comment 7: Why isn't a mixed-effects model applied in interpreting the results, especially considering the non-independence of repeated measures across different AEZs and years?
Response 7: We think that samples were independent between sampling date, agroecological zones, season and year. And in each agroecological we could consider as randomized block design and performed analysis independently. However, GLM fit well when considering variable replicates number across seasons and years.
Comment 8: The discussion section is the weakest point of the paper and it does not manage to justify the whole concept of the research. While the study involved agroclimatic factors such as rainfall, temperature, and humidity, further exploration of other biotic factors (e.g., natural predators or crop varieties, hybrids) would be valuable in understanding the broader ecological implications of such an intricate ecosystem.
Response 8:
We have revised the Discussion section to strengthen the justification and address these gaps. Specifically, we now discuss the influence of biotic factors such as natural enemies, maize varieties and agricultural practices in shaping FAW dynamics. While these were not directly measured in our study, we refer to relevant studies that highlight their role.
Comment 9: Even though the variation between years is addressed, the paper does not discuss why this variation occurred
Response 10: We now provide a more detailed explanation of year-to-year variations.
Comment 11: The argumentation of drowning larvae is pure speculation or bare local observations.
Response 11: Regarding the observation of drowning larvae, we acknowledge that this was based on field observations.
Comment 12: How did the study and the results of the spatial distribution of FAW larvae across fields contribute to improving control methods?
Response 12: We have revised the Discussion and Conclusion sections to clarify how the spatial distribution results can inform pest management strategies. The moderate aggregation of FAW larvae (δ = 1.25) supports the development of targeted sampling plans and site-specific control interventions.
Comment 13: Based on which part of the study the authors could conclude that proactive practices such as intercropping, optimized planting times, and trap cropping could be implemented to disrupt FAW oviposition and larval feeding?
Response 13: Intercropping, optimized planting times, and trap cropping can’t be justified by my study. They are discussed as recommendations based on the observed FAW behavior and findings from similar studies in Africa and Latin America. This part is removed from discussion section.
Comment 14: There is no information regarding maize hybrids or other cultivation practices (e.g., BT crops).
Response 14: We have provided information on maize hybrids or other cultivation practices in the discussion section.
Comment 15: Although a significant amount of work appears to have been invested in this study, the results do not seem to provide conclusive answers to the main research questions posed in the paper, given the complexity of the topic
Response 15: This point has been clarified in the Conclusion.

Round 2
Reviewer 2 Report
Comments and Suggestions for Authors
The authors have addressed many of the suggestions that I mentioned in the previous review, but important issues still remain.
1. The concept of a "basic colony" in the Abstract will not be understood by a normal reader. Please explain this better. Do you mean that on average you observed two larvae per plant?
2. (L152-153) Why have you changed the area of fields to Km2? Agricultural fields are measured in hectares.
3. My previous request for information on the error structure and link function used in the GLM models has NOT been addressed. (previous comment 16)
4. I previously requested treatment and error degrees of freedom for F statistics. The error df values have not been given for F statistics in Table 2, Table 4, Table 6 or Table 8.
5. What does "Rep" mean in Table 2, Table 4, Table 6 and Table 8? Please use a complete word or explain.
6. L437 should read "...more foliar damage during the dry season".
7. There are numerous formatting issues in the References section.
8. The authors should check their manuscript for typos (I spotted quite a lot)
Author Response
RESPONSES TO REVIEWER 2 COMMENTS
Dear Editor
We would like first to thank you and the reviewer for your contributions to improving the quality of our manuscript. We proceeded question by question. Changes are in red in the revised version of the manuscript.
Comment 1: The concept of a "basic colony" in the Abstract will not be understood by a normal reader. Please explain this better. Do you mean that on average you observed two larvae per plant?
Response 1: By "basic colony," we mean "initial cluster" in the study. We gave explanation in the Abstract.
Comment 2: (L152-153) Why have you changed the area of fields to Km2? Agricultural fields are measured in hectares.
Response 2: This change was asked we by the previous reviewer 1. We revised now the text to revert to hectares (ha) to maintain consistency with standard agricultural reporting.
Comment 3: My previous request for information on the error structure and link function used in the GLM models has NOT been addressed. (previous comment 16)
Response 3: We used GLM procedure to perform different comparisons because sampling was independent when considering farms sampled in each agro-ecological zone, season or year with variable replicate number. And each season or year could be considered as a randomized block design with variable replicate number.
In table 3, we compared the larval infestation in the two agroecological zones across the the two years for each season and error structure could be expressed as following:
Yijk = μ + Tij + Ï“k +eijk
Where Y is larval infestation rate
μ is the overall mean
Tij is the effect of zone i (i=1, 2) and of the year j (j=1, 2)
Ï“k is the effect of replicate k (k=1, 2, 3, ….replicates)
and eijk is the residual error
In table 5 we compared larval density between weeks for each year and each season and error structure was:
Yij = μ + Tij + Ï“k +eijk
Where Y is larval density
μ is the overall mean
Tij is the effect of week i (i=1, 2, 3, 4, 5) and of the zone j (j=1, 2)
Ï“k is the effect of replicate k (k=1, 2, 3, ….replicate number)
and eijk is the residual error
In table 7, we compared the percentage of damaged plants between agroecological zones across years or across season the model was:
Yij = μ + Tij + Ï“k +eijk
Where Y is the percentage of damaged plants
μ is the overall mean
Tij is the effect of season or year (i=1, 2) and of the zone j (j=1, 2)
Ï“k is the effect of replicate k (k=1, 2, 3, ….replicate number)
and eijk is the residual error
In Table 9, we compared the weekly damaged leaf for each season within each year, and error structure was:
Yij = μ + Tij + Ï“k +eijk
Where Y is larval density
μ is the overall mean
Tij is the effect of week i (i=1, 2, 3, 4, 5) and of the zone j (j=1, 2)
Ï“k is the effect of replicate k (k=1, 2, 3, ….replicate number)
and eijk is the residual error
Comment 4: I previously requested treatment and error degrees of freedom for F statistics. The error df values have not been given for F statistics in Table 2, Table 4, Table 6 or Table 8.
Response 4: We revised Tables 2, 4, 6, and 8 and include the error degrees of freedom alongside the treatment df in reporting F statistics.
Comment 5: What does "Rep" mean in Table 2, Table 4, Table 6 and Table 8? Please use a complete word or explain
Response 5: The term "Rep" refers to "Replicates" in the experimental design. We modified it in the table captions.
Comment 6: L437 should read "...more foliar damage during the dry season"
Response 6: We revised the text to read: "...more foliar damage during the dry season."
Comment 7: Formatting issues in References
Response 7: The References section was revised thoroughly, and all formatting inconsistencies were corrected to align with the journal's guidelines.
Comment 8: The authors should check their manuscript for typos (I spotted quite a lot)
Response 8: The entire manuscript was rechecked carefully proofread to address any remaining typos or inconsistencies.

Round 3
Reviewer 2 Report
Comments and Suggestions for Authors
The authors have greatly improved their manuscript. I spotted just a few errors and details that need correcting prior to publication.
Abstract: L37. Reword to: ...with a basic colony of 2.08 larvae, i.e., an average initial cluster of 2.08 larvae observed per plant....
L179. The F values should follow the factor rather than being listed at the end of the sentence. For example: This analysis reveals that larval infestation rates vary significantly according to zone (F1,317 = 4.35; p < 0.03), season (F1,317 = 48.20; p < 0.0001), and year (F1,317 = 8.07; p < 0.004).
*This needs to be modified in the entire results section*.
L397. Change to "For example, Fiaboe et al. [62] recently observed...
L404 change to "Hailu et al. [65] found that intercropping maize
L415. Change "the America" to "the United States".
L417. Change evolve to "establish"
Ref [10] article number is missing. Should be Front. Plant Sci. 2023, 13, 1079442.
Ref [13] article number is missing. Should be J. Integr. Pest Man. 2020, 11(1): 20.
Refs [15], and [33], article number is missing.
Ref [74] Journal name is missing – should be J Agric. Sci. Technol.